# The Relationship between Environmental Regulation, Industrial Transformation Change and Urban Low-Carbon Development: Evidence from 282 Cities in China

**DOI:** 10.3390/ijerph191912837

**Published:** 2022-10-07

**Authors:** Kun Chen, Yinrong Chen, Qingying Zhu, Min Liu

**Affiliations:** 1College of Public Administration, Huazhong Agricultural University, Wuhan 430700, China; 2School of Public Administration, South China Agricultural University, Guangzhou 510642, China

**Keywords:** urban low-carbon development, environmental regulation, industrial structure, optimization, upgrading, spatial panel Durbin model, panel threshold model

## Abstract

Environmental regulation (ER) plays an important role in urban low-carbon development (ULCD). First of all, we evaluate the ULCD level of 282 cities in China from 2005 to 2020 by constructing an index group and entropy method. Two panel models are then used to test the spillover effects and threshold effects of ER and industrial structure on ULCD. The results show that the ULCD level of most cities is still in grade III (0.27–0.38) or IV (0.38–0.49), and the level of central-western cities is generally lower than that of eastern cities. Furthermore, the spillover effect of ER and industrial structure upgrading (UIS) on ULCD is positive in eastern cities (0.038) but opposite in central or western cities (−0.024). Further results show that the positive effects of optimization of industrial structure (OIS) and UIS are gradually increasing with the improvement of ER. However, the positive effects are more beneficial to the eastern cities. Therefore, the conclusions of this study can provide a decision-making reference for local government to comprehensively formulate environmental and industrial policies to enhance the low-carbon development of cities.

## 1. Introduction

Domestic carbon dioxide emissions should peak by around 2030, and achieving carbon neutrality by 2060 is one ambitious goal officially put forward by China in 2020. Due to excessive emphasis on economic growth, in recent decades the economic development led by the Chinese government has led to serious eco-environmental problems and large emissions of CO_2_ [1,2]. China has been vigorously promoting green development, actively participating in carbon emission reduction actions, and unswervingly following the path of low-carbon development [3]. Though there is no unified standard for the concept of urban low-carbon development (ULCD) in the world, many scholars agree that low-carbon development requires not only carbon emission reduction but also healthy economic development [4]. The city is the main space carrier of human activities and is a complex system composed of economy, society, resources, environment, and other subsystems. The ULCD is a complex process involving economic and social development, technology accumulation, energy consumption, and environmental protection [5,6]. The core of it is to reduce carbon emissions while considering the stable operation of the social and economic system. From the perspective of measures to reduce carbon emissions, environmental regulation (ER) is a relatively common and proven effective administrative intervention adopted by governments [7,8].

From the content of the existing research, more and more scholars have set out from the whole urban system to build a multi-dimensional index group to measure the comprehensive level of ULCD [4,9]. However, direct research on ER and ULCD is rare, and many studies mainly focus on limited aspects [7,10,11]. Some of the relevant conclusions are not completely consistent. Liu and Xie used the industry panel data model to prove that ER will restrain the independent innovation ability of enterprises and chose technology introduction under the influence of cost effect [11,12]. However, some scholars have also found that ER can promote technological innovation, further promote industrial upgrading, improve energy efficiency, and reduce pollutant emissions [7,13]. Nevertheless, this incentive effect has significant characteristics of spatio-temporal heterogeneity, which is more beneficial to developed areas [14]. Due to the non-synchronization of social and economic development stage, the pressure of ER is significantly different in space. This difference makes polluting enterprises transfer to areas with backward economic development and weak environmental regulation [11,15]. Although the weak environmental regulation in the backward areas will attract polluting enterprises at a certain stage, the transfer of such enterprises has brought economic growth and enhanced comprehensive development capacity to underdeveloped areas. It ultimately provides a driving force for the birth of stronger environmental regulation [16]. Many studies have discussed the role of ER as a certain aspect of low-carbon development, and limited studies have analyzed the spillover or non-linear effects of ER. The work of studying the effect of ER from the comprehensive perspective of low-carbon development is not rich, which may blur the understanding of the law of ULCD.

The most important premise of all is to evaluate the level of ULCD scientifically. Based on existing research, we construct the evaluation index group of ULCD and use the entropy method to calculate the level of 282 cities from 2005 to 2020. Furthermore, we use the panel regression method to briefly examine the relationships between variables. Considering the impact of ER on the production behavior of polluting enterprises, on the one hand, ER may lead to the spatial transfer of industry, technology, and labor, and ultimately affect the ULCD [17,18]. On the other hand, the impact of ER on the social economy may depend on the intensity of ER adopted by local governments and show a non-linear relationship [16,19]. Therefore, we tend to utilize the spatial panel model to investigate the spillover effects of ER and industrial structure on ULCD. Finally, the panel threshold model tests the non-linear relationship between ER, IS, and ULCD. The conclusion of this study is expected to provide references for local governments to comprehensively formulate environmental regulation and low-carbon development policies.

## 2. Literature Review

The early studies on the evaluation of ULCD mostly focus on the absolute reduction of CO_2_ emissions, and the relevant assessments are usually reflected by one or more indicators. With the in-depth study of low-carbon development, scholars realize that the research conclusions and policy recommendations based on a small number of indicators to characterize ULCD cannot meet the actual needs of managers and believe that ULCD is an orderly, coordinated, and sustainable way of development [4,9]. On this basis, a wealth of scholars have devoted themselves to the construction of an index group to measure the ULCD from the perspective of a social–economic–ecological integrated system [20,21]. As for research scale, some scholars mainly estimate the differences in low-carbon development level in regions on a national scale or provincial scale, and there are few studies on the urban scale, but the research on the urban scale have been gradually enriched in recent years [7,12,13]. As for the research methods, they are mainly divided into single-index and multi-index evaluation methods. The single-index method mainly uses CO_2_ emissions to characterize ULCD, which is relatively simple. Multi-index evaluation methods are diverse due to the differences in the treatment of indicators, such as the TOPSIS method, the Analytic hierarchy process, entropy weight method, and so on [11,16,22]. Generally speaking, current research on low-carbon development assessment is very rich, and the evaluation methods and index systems are relatively mature, but the research regarding long-time series of ULCD across the country is rare.

In recent decades, people have gradually realized that greenhouse gas emissions such as CO_2_ will seriously impact the global climate. Therefore, many environmental regulations have been formulated and adopted to reduce the global emission of pollutants [23]. Academia has carried out a wealth of theoretical and empirical research on environmental regulation. The present research results show that there are three hot spots of universal concern, one of which is the environmental Kuznets curve. Grossman and Krueger found an inverse U-shaped curve between pollution emission levels and income, which is called the environmental Kuznets curve [24]. Since then, many scholars have empirically tested the existence and inflexion point of the environmental Kuznets curve [25,26]. Secondly, the Porter hypothesis is another topic of common concern. Porter and Van der Linde believe that strict ER will lead to technological innovation, which will enhance productivity and market competitiveness [27]. However, some studies believe that industries with more stringent ER may have a decline in competitiveness [10,28]. The last one is the “pollution paradise hypothesis”. Its core view is that when a country strengthens environmental regulation policies, the polluting enterprises of this country will move to other countries with weak environmental regulation policies, and those countries provide refuge for polluting enterprises [29]. With the research’s deepening, the attention of scholars has gradually shifted from enterprises between countries to enterprises between different regions of a country. Although these theories have been questioned, the environmental Kuznets curve, the Porter hypothesis, and the pollution paradise hypothesis still provide important theoretical references for the follow-up research on environmental regulation.

As for China, under the tremendous pressure of global climate change and carbon emission reduction targets, the intensity of environmental regulation is increasing yearly. China tries to guide cities to achieve low-carbon and green development by employing ER, but how does ER promote ULCD? According to theoretical and empirical studies, ER will impact the industrial behavior of enterprises. With the increase in the degree of ER, it will first increase the production cost of polluting enterprises. As stated in the Porter hypothesis, ER can promote enterprises’ technological development and reduce environmental pollution’s negative externalities by imposing emission reduction constraints. Furthermore, polluting enterprises in developed cities are transferred to less developed cities. Environmental regulation causes the cross-regional and inter-city transfer of polluting enterprises and industries. For the impact of central or western cities, according to the environmental Kuznets curve hypothesis, before reaching the “inflection point”, the economic benefits brought by polluting enterprises are greater than the environmental costs, thus promoting regional development. Overall, ER has a non-linear effect on the low-carbon development of social, economic, and ecological environments in different regions and cities and finally shapes the differentiated urban low-carbon development pattern.

## 3. Materials and Methods

### 3.1. Variable Setting and Calculation

#### 3.1.1. Dependent Variable

Urban low carbon development (ULCD): As this paper mainly studies the development level of an urban low-carbon economy, ULCD indicators mainly consider the content closely related to the city; drawing lessons from previous studies, we have found that urban economy, society, and the ecological environment are inevitable aspects to be considered. In addition, the use of energy resources is directly related to the level of ULCD, and the level of urban resource utilization must be taken into account in assessing the level of ULCD. The differences in the development of different cities are closely related to government-led urban development planning, so the evaluation of ULCD also takes into account the factors of urban planning. Finally, based on the connotations of ULCD, we construct an evaluation index group from five dimensions, including low-carbon economy, low-carbon society, resource utilization, urban planning, and low-carbon environments, to evaluate the ULCD level of each city [4].

To avoid subjectivity, the weights of indexes are obtained by the entropy method, and the results are shown in Table 1. Formula (A1)–(A7) shows the specific steps.

#### 3.1.2. Independent Variables

Environmental regulation (ER): Referring to Du et al., we comprehensively evaluate the degree of ER by using industrial dust removal rate, industrial SO_2_ removal rate, the comprehensive utilization rate of general industrial solid waste, harmless treatment rate of domestic waste, and centralized sewage treatment rate [14]. The reason for this is that, compared with environmental regulation policies, environmental supervision frequency and other environmental regulation indicators, they are statistically universal and easy to obtain. Secondly, these indicators can better reflect the effect of urban environmental regulation and can meet the needs of this study. Similar to the calculation of the explanatory variable, the index of environmental regulation is also obtained by the entropy method. The greater the environmental regulation index, the greater the pressure on polluting enterprises.

Industrial structure: Referring to Zhao et al., in this paper the industrial structure is divided into industrial structure optimization (OIS) and industrial structure upgrading (UIS) [30]. The UIS is represented by the ratio of the tertiary industry output value to the secondary industry output value. The OIS is represented by the Theil index, which is a negative indicator [30]. The Theil index is calculated as Formula (1).
(1)$$OIS=\sum_{i=1}^{n}(\frac{Y_{i}}{Y})\ln(\frac{Y_{i}}{L_{i}}/\frac{Y}{L})$$
where, *i* is the *i*th industry; *n* is the number of industries, with a maximum of 3; *Y* is the total output value; *L* represents the total employment; *Y_i_* depicts the output value of the *i*th industry; and *L_i_* is the number of employees in *i*th industry.

#### 3.1.3. Control Variables

The control variables include education level (edu), investment level (ifa), science and technology level (tec), and economic openness (eol), with reference to the relevant research in this paper [4,14,30]. Among them, the indicator of education level is the number of senior high school graduates and above. The index of investment level is the amount of investment in fixed assets. The index of the level of science and technology is the expenditure on science and technology development of government. The index of economic openness is the foreign capital utilized by the city.

### 3.2. Methods

#### 3.2.1. Benchmark

The panel regression model is selected as the benchmark model due to the characteristics of the data. Because of the heteroscedasticity of the data, the variables are processed in the form of a natural logarithm. The general panel regression model of this study is constructed as follows:(2)$$\begin{array}{ll} ulcd_{it} & =a_{i}+\lambda_{t}+\beta_{1}er+\beta_{2}ois+\beta_{3}uis+\beta_{4}edu\\ & +\beta_{5}ifa+\beta_{6}tec+\beta_{7}eol+\varepsilon_{it} \end{array}$$
where *ulcd_it_* is the urban low-carbon development level of the *i*th city in the *t* year; *α_i_* represents the individual effect; *λ_t_* represents the time effect; *ε_it_* is the error term; *β* is the parameter to be estimated; *er* depicts environmental regulation; *ois* is the optimization of industrial structure; *uis* is the upgrading of industrial structure; *edu* represents the level of education; *ifa* represents the level of investment; *tec* indicates the level of science and technology; and *eol* indicates the degree of openness of the economy.

#### 3.2.2. Spatial Panel Durbin Model (SPDM)

The results of the benchmark regression model can preliminarily reveal the relationship between the ER, industrial structure, and ULCD of each city, but the ability to analyze the effect of ER in neighboring cities on the level of ULCD is insufficient. Thus, we further use the spatial panel model to analyze the spatial impact of ER and industrial structure on ULCD. As for model selection, starting with the Durbin model provides a better choice [31]. The SPDM includes the spatial lag item and the spatial error item shown in Formula (3).
(3)$$ulcd_{it}=\rho W*lcd_{it}+\beta_{n}M_{it}+\gamma W{\bar{M}}_{it}+\theta W\varepsilon_{it}+e_{it}+\mu_{i}+\lambda_{it}$$
where *ρ* and *θ* represent the spatial effect coefficient; *β_n_* indicates the coefficients of *k* explanatory variables; *W* denotes the spatial weight matrix, which is obtained by anti-geographical distance; *M* is a group of explanatory variables, including ER, OIS, UIS, and other control variables; $\bar{M}$ is a matrix of independent variables; *W*$\bar{M}$*_it_* reflects the values of spatial lag explanatory variables in adjacent regions; *γ* is used to measure the marginal influence of the explanatory variables in the adjacent regions on the explained variables; *ε* represents the spatial autocorrelation error term; *e_it_* is a random disturbance term; *μ_i_* represents the spatial fixed effect; and *λ_t_* denotes the time fixed effect. 

#### 3.2.3. Panel Threshold Model (PTM)

The intensity of ER is dynamic, and its impact on ULCD may have significant phased differences and finally show non-linear characteristics. As for the non-linear characteristics of the influence effect, this study uses the PTM to verify it, as per Cui et al. [32]. The panel threshold model of this study is set as Formula (4).
(4)$$y_{it}=\delta_{1}x_{it}I(q_{it}\leq \gamma )+\delta_{2}x_{it}I(q_{it}>\gamma )+\mu_{i}+\varepsilon_{it}$$
where *y_it_* is the interpreted variable; *δ* is the coefficient of the explanatory variable *x_it_* to the explained variable when the threshold variable, *q_it_*, is greater than or less than the threshold *γ*; and *I* represents the indicator variable. When *q_it_* ≤ *γ*, *I* = 1; Otherwise, *I* = 0. *μ_i_* is the individual effect of the region and *ε_it_* is the error term. After derivation, the single threshold model (5) and double threshold model (6) can be obtained.
(5)$$ulcd_{it}=\delta_{0}+\delta_{1}x_{it}I(er_{it}\leq \gamma_{1})+\delta_{2}x_{it}I(er_{it}>\gamma )+\beta_{n}N_{it}+\mu_{i}+\varepsilon_{it}$$
(6)$$\begin{array}{ll} ulcd_{it} & =\delta_{0}+\delta_{1}x_{it}I(er_{it}\leq \gamma_{1})+\delta_{2}x_{it}I(\gamma_{1}<er_{it}\leq \gamma_{2})\\ & +\delta_{3}x_{it}I(er_{it}>\gamma_{3})+\beta_{n}N_{it}+\mu_{i}+\varepsilon_{it} \end{array}$$
where *er_it_* is the threshold variable; *x_it_* is the core explanatory variable, including *ois* and *uis*; and *β_n_* is the coefficient of the control variables.

### 3.3. Data

The main data of 282 cities from 2005 to 2020 include low-carbon development evaluation data, environmental regulation data, industrial structure data, and other control variables’ data. These data come from China Urban Statistical Yearbooks (2006–2021) and provincial statistical yearbooks (2006–2021). The indexes of ULCD are shown in Table 1 [4,15,30]. Before the evaluation, the indexes are standardized. In addition, to further improve the accuracy and comparability, the value indicators are measured by the deflator method up to 2005. Furthermore, in this paper the 282 cities are divided into 86 eastern cities and 196 central-western cities. The eastern region includes Beijing, Tianjin, Hebei, Shanghai, Jiangsu, Zhejiang, Fujian, Shandong, Guangdong, and Hainan. Other provinces and cities belong to the central-western region. It should be noted that, due to lack of data, Xinjiang, Tibet, Hong Kong, Macao, Taiwan, and parts of cities are not within the scope in this study.

## 4. Results

### 4.1. Urban Low-Carbon Development Level

According to the evaluation index group and entropy weight method, the ULCD level of 282 cities from 2005 to 2020 is obtained, and the results are shown in Figure 1. For convenience, this paper uses the results of 2005, 2008, 2011, 2014, 2017, and 2020 as examples and uses the map display and grading color tool of ArcGIS10.3 software to divide the results of ULCD level into six grades. The higher the grade, the higher the level of ULCD. As can be seen from Figure 1, the ULCD level of the observed sample cities shows an overall upward trend over time, but even in 2020, the ULCD level of most cities is still in grade III or IV, indicating that the comprehensive level of ULCD is not high. In an east-west direction, the ULCD level in the eastern region is the highest, followed by the central region and the western region. The reason for this situation may be that the input level of labor, capital, and technology in the eastern region is significantly higher than that in other areas since the Reform and Opening up, resulting in industrial structure adjustment and technological progress significantly raising the level of urban ULCD. As the central region is closer to the eastern region, the development of the eastern region gives priority to the industrial and technological development of the central region. In the north-south direction, the level of ULCD in the north is significantly higher than that in the south, which may be due to the fact that cities in the north have convenient topographical conditions and rich natural resources compared with those in the south, resulting in more rapid economic development and more convenient industrial adjustment and technological exchanges. In addition, Beijing, Shanghai, and Chongqing have become high-value regional low-carbon development centers.

### 4.2. Benchmark Result 

After the evaluation of ULCD, we further use the panel regression model to analyze the impact of ER and industrial structure on ULCD. In order to draw a more accurate conclusion, LM test and Hausman test are used to select the ideal panel model. The final results are shown in Table 2.

From the results of the LM test and the Hausman test in Table 2, we can see that we should adopt the “fixed effect model”. For convenience and simplicity, only the results of the fixed effect model are shown in Table 1. The benchmark results show that the coefficients of ER (“*ln er*”) in different models are significant, and they are 0.4795, 0.516, and 0.205, respectively, which shows that ER has a significant positive effect on ULCD. In addition, education level (“*ln edu*”), economic development level (“*ln pdgp*”), and economic openness (“*ln eol*”) are all statistically significant in the three models, and their coefficients are all positive. The difference is that the coefficient of OIS (“*ln ois*”) only passed the test in the eastern cities, and it is −0.221, but the coefficient of UIS (“*ln uis*”) only passed the test in the central-western cities, and the coefficient is −0.541. We can see that the unbalanced industrial structure limits the ULCD in eastern cities, and the OIS of central or western cities suppresses the ULCD during the research period. In addition, the level of scientific and technological development (“*ln tec*”) only promotes the ULCD in eastern cities.

### 4.3. Spatial Spillover Effect

In this paper, the global Moran’ I is used to detect the existence of the spatial effects of the main factors based on the inverse distance weight. The results of global Moran’ I in typical years are shown in Table 3, where there are significant spatial correlation features of the level of the main variables.

Furthermore, based on the samples of eastern cities and central-western cities, we carried out LM-lag, Robust LM-lag, LM-error, and Robust LM-error tests on the model (3), and all items passed the 5% significance test, indicating that there are lag terms and error terms in the spatial dependence of ULCD, and SPDM should be selected. Secondly, both Wald and LR tests pass the significance test of 1%, indicating that they highly reject the hypothesis that SPDM can be simplified to PSEM or PSLM. Finally, the fixed effect model is selected by the Hausmann test. Through the SPDM analysis of different fixed effects, it is found that the values of *R*^2^ and logarithmic likelihood functions in the individual fixed effect model are relatively larger, which shows that the ideal model is the individual fixed one. The results of the individual fixed model are shown in Table 4.

According to *SPDM* results of eastern cities and central-western cities in Table 4, except for the lag items of OIS (*W*×*n ois*) in central-western cities, the lag coefficients of other variables are significantly not equal to zero, which confirms that ER, OIS, and UIS have spatial spillover effects on ULCD. The direct, indirect, and total effects are then further decomposed by the method introduced by Lesage and Pace. The indirect effect corresponds to the spillover effect, and the results can be seen in the *Indirect* column in Table 4. 

The results of spillover effects show that the spillover effect coefficients of ER in the eastern and central-western cities are 0.038 and −0.024, respectively, both of which are significant at a 95% confidence interval. It is shown that the ER has a positive spillover effect on the ULCD of eastern cities, but has the opposite effect for central-western cities. The spillover effect of OIS is only significant in the eastern cities, and it hurts the ULCD. The spillover effect coefficients of UIS in the eastern and central-western cities are 0.063 and −0.042, respectively, indicating that, in the eastern region, the UIS of the city has a positive impact on its ULCD and the ULCD of neighboring cities. However, it is harmful to the ULCD of the surrounding cities in the central-western region.

### 4.4. Threshold Effect

Taking the ULCD level as the explained variable, we estimate whether there is no threshold, one threshold, or two thresholds for ER in eastern and central-western cities. Referring to the bootstrap method, we use State15 statistical software to obtain the corresponding *p* value through repeated sampling 1000 times to judge whether there is a threshold effect. The results are shown in Table 5.

As seen from Table 5 above, when ER is a threshold variable, F statistics fail to assume that there are three thresholds in the two models. Therefore, both models have two threshold values, and the estimated results of threshold values are given in Table 6. Furthermore, the authenticity of the threshold can be reflected by likelihood ratio statistics. Due to the space limitation, the likelihood ratio statistics diagrams are shown in Figure A1.

After verifying the authenticity of the threshold, we can further analyze the threshold effect. According to the results of the eastern cities in Table 7, when *er* ≤ 0.5361, the coefficient of OIS is −0.3912 and passes the significance test, while the coefficient of UIS does not pass the significance test. When 0.5361 < *er* ≤ 0.8110, the coefficient of UIS is −0.0135, while the coefficient of UIS is 0.1739, and both have passed the significance test. When 0.8110 < *er*, the coefficient of OIS is very small and does not pass the significance test, while the coefficient of UIS passes the significance test and is 0.1938. The above results show that under an increasing state of ER, the bondage effect of OIS gradually disappears, but the promotion effect of UIS is gradually enhanced.

From the perspective of central-western cities, when *er* ≤ 0.2200, the coefficient of OIS is −0.4309, and that of UIS is −0.1147, both of which are significant. When 0.2200 < *er* ≤ 0.5834, the coefficient of OIS is −0.0740, while that of UIS is −0.0831, and both have passed the significance test. When 0.5834 < *er*, the coefficient of OIS is 0.1362 and passes the significance test, while the coefficient of UIS is contrary. The results show that when the intensity of ER in central-western cities is weak, the impact of OIS on ULCD is negative. However, with the increase of the intensity of ER, the negative impact gradually weakens. Finally, the negative impact gradually turns into a positive impact. In addition, similar to the results of OIS, with the improvement of the intensity of ER, the negative impact of UIS gradually weakens, but it does not reverse to positive in the end.

### 4.5. Robust Test

Per capita CO_2_ emissions in urban economic growth are generally regarded as an important symbol of low-carbon development level. Therefore, we replace the explained variables with CO_2_ emissions per unit of GDP (*ln pgCO_2_*) into the benchmark model to further verify the robustness of the results. It should be noted that CO_2_ emissions are obtained based on the energy consumption of coal, coke, crude oil, fuel oil, gasoline, diesel, natural gas, and their respective CO_2_ emission coefficients. The energy consumption data come from the 2006–2021 Provincial Statistical Yearbooks and the Urban Statistical Yearbooks. The CO_2_ emission coefficients come from the IPCC report [33]. In order to be consistent with the above model, only the individual fixed effect model is discussed here. Due to the replacement of the explained variables, we have carried out the necessary tests, and the process is shown in Table A1, Table A2 and Table A3 in Appendix A. The final results of robust test models are shown in Table 8.

According to the benchmark model (*BM*) results in Table 8, the coefficients of ER in eastern cities or central-western cities are negative. That is, with the increase in ER intensity, CO_2_ emissions per unit of GDP show a downward trend, reflecting the improvement of ULCD. The impact of OIS and UIS on CO_2_ emissions per unit of GDP of eastern or central-western cities are similar to the initial results. That is, OIS increases the CO_2_ emissions of eastern cities, which is not conducive to the ULCD. The impact of UIS on CO_2_ emissions per unit of GDP in central-western cities is also similar to the initial result. From the perspective of spillover effects, due to the space limitation, the robustness analysis only discusses the existence of spillover effects. According to the *SPDM* results of eastern and central-western cities, the spatial lag coefficients of the variables in the two models are significantly not equal to zero, which proves that the spillover effect does exist.

As regards the threshold effect (*PTM*), the two threshold values of eastern cities in Table 8 are smaller than those of the original model. In comparison, the two threshold values of central-western cities are larger than those of the original model. With the increase in the intensity of ER, the impact of OIS on CO_2_ emissions per unit of GDP in eastern and central-western cities is positive but gradually decreases. When the level of ER changes from low to medium, the impact of UIS on CO_2_ emissions per unit of GDP in eastern cities changes from positive to negative, but when the ER level is high, the impact is insignificant. When the ER level is medium or below, the impact of UIS on CO_2_ emissions per unit of GDP in central-western cities has always been positive, but when at the high ER level, the UIS in central-western cities will significantly reduce CO_2_ emissions per unit of GDP. Generally speaking, the results of the threshold effect in Table 8 are consistent with those of the original model. Part of the difference lies in the size of the threshold and the significance of the impact of OIS and UIS under high-intensity ER. The reason for this may be the research errors caused by the acquisition of the two explained variables and the differences in data sources, but the robustness of the research results and conclusions cannot be denied.

## 5. Discussion

This paper mainly focuses on the impact of ER, OIS, and UIS on ULCD in different cities in China. Although the ULCD level of Chinese cities is increasing, the overall level is still in the middle state, and there are obvious spatially unbalanced characteristics. It is consistent with the conclusion of Wang et al. that the quality of low-carbon development in the eastern region is better than that in the central and western regions [4]. However, the difference is that we found the level of urban low-carbon development has a significant spatial correlation, and Beijing, Shanghai, and Chongqing have become the leaders of the surrounding cities. In addition, Wang et al. believe that the transformation and upgrading of industries can improve the level of ULCD, but we found in this study that OIS or UIS shows a significant negative impact on ULCD during the period. Therefore, the heterogeneous impacts of ER, OIS, and UIS are worthy of in-depth analysis.

Further research found that no matter which explanatory variable is used, all the evidence supports that ER is conducive to ULCD, which is close to the conclusions of Pei et al. [34]. The reason is that increasing environmental regulation intensity can eliminate backward industries, improve technical efficiency, and reduce CO_2_ emissions. However, from the perspective of the spillover effect, we find that the impact of ER in eastern cities is positive, while that in central-western cities is negative. The explanation may be found from the study of Zhao et al., that is, the comprehensive level of development and the intensity of ER in the eastern region are higher than those in the central and western regions, resulting in greater restrictions on carbon-intensive industries in the eastern region, while the opposite is true in the central and western region [35]. This study also provides some support for such an explanation; that is, the UIS has a positive spillover effect on the ULCD of eastern cities, while it has a negative spillover effect on the ULCD of central or western cities. The difference is that we find the OIS’s spillover effect negatively impacts the ULCD of eastern cities.

The threshold results show that with the continuous improvement of the ER level, the inhibitory effect of OIS on urban low-carbon development in eastern cities gradually disappears, but the UIS promotes the ULCD level gradually. However, the impacts of OIS and UIS on the ULCD in central-western cities are always negative, but the negative influences become weaker and weaker. The reason for this may be that the government tended to give priority to economic development rather than ecological environment protection in the early stage, which led to the non-exclusion of carbon-intensive industries in the process of OIS. Due to the location advantage, the secondary and tertiary industries in the eastern cities developed rapidly. With the steady growth of the economy, the problem of the urban ecological environment in the eastern region has become increasingly prominent. The eastern region forces the transformation of industrial structure and the transfer of carbon-intensive industries through more stringent environmental regulations, which comprehensively leads to the impact of OIS and UIS on ULCD. Because the development of the central and western regions lags behind that of the eastern regions, the central and western regions are forced to undertake the industrial transfer of the cities in the eastern region, which comprehensively leads to the change of industrial structure, but increases the social and economic carbon dioxide emissions of the central or western cities.

Finally, there are some shortcomings in this paper. First of all, although this paper’s relevant results and conclusions have been tested for robustness, there are some missing samples in the central-western cities due to the difficulties in obtaining data. As a result, the results and conclusions about the central-western cities can only partially reflect their regular characteristics. Secondly, some studies believe there may be a time lag in the impact of ER, OIS, and UIS on the social economy. Therefore, in further research, we should strengthen the ability to obtain data and consider improving the model to analyze the time lag of the impact of the main variables.

## 6. Conclusions

This study comprehensively evaluates the low-carbon development level of 282 cities in China by using economic, social, resource, planning, and ecological indicators. Econometric panel models are then used to analyze the linear, non-linear, and spatial spillover effects of ER, OIS, and UIS on ULCD. The conclusions are as follows.

From 2005 to 2020, most Chinese cities have been committed to improving the level of ULCD and achieved remarkable results, but the overall level of ULCD is still not high and is characterized by spatial imbalance. The linear effect concludes that ER shows a significant positive impact on ULCD. At the same time, government-led industrial structure optimization and upgrading have little influence on the improvement of ULCD during the period. Thus, the eastern and central-western cities should adopt diversified policies and measures to improve the overall level of ER. Furthermore, when promoting the transformation of industrial structure, the government should strengthen the identification and selection of industries, raise the entry threshold for carbon-intensive industries, and prioritize introducing high-tech industries.

The result of the spatial spillover effect reveals that strengthening the ER and UIS of cities in the eastern region can improve the ULCD levels of other neighboring cities, while the central-western cities are the contrary. The OIS of a city in the eastern region may limit the ULCD of surrounding cities, but it is not significant in the central and western regions. The results of the threshold effect show that with the intensity of ER from small to large, the negative impact of OIS on the ULCD of eastern and central-western cities gradually weakens and even changes to positive. However, the UIS of eastern cities plays a more and more important role in promoting the ULCD, and the negative impact of UIS on the ULCD of central-western cities gradually weakens. From a comprehensive point of view, the influence of ER on ULCD must be a long-term and non-linear process. The municipal governments of the eastern and central-western cities should issue long-term effective environmental policies and measures and carry out a periodic evaluation of the performance of these regulation instruments to adjust the intensity of ER in time, which ensures that environmental policies can always have a positive impact on ULCD.

## Figures and Tables

**Figure 1 ijerph-19-12837-f001:**
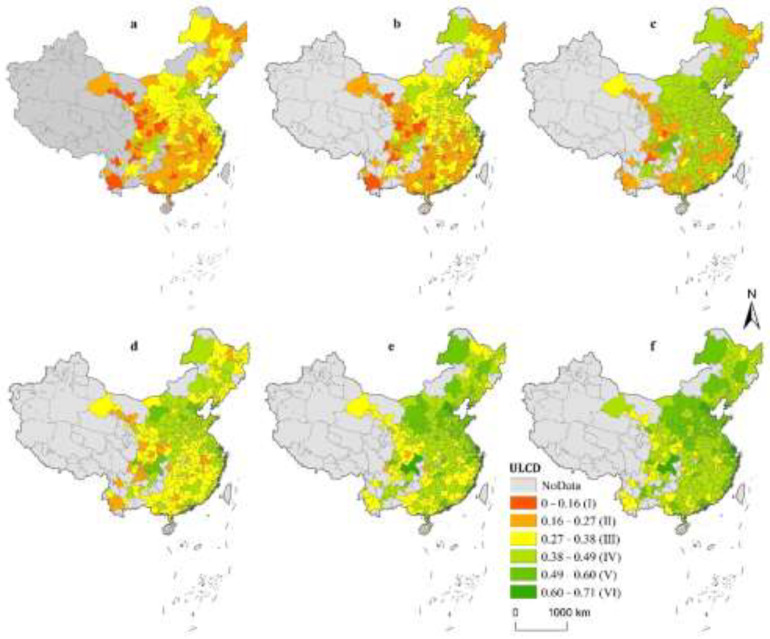
Spatio-temporal characteristics of the ULCD level of 282 cities in China ((**a**): 2004; (**b**): 2008; (**c**): 2011; (**d**): 2014; (**e**): 2017; (**f**): 2020).

**Table 1 ijerph-19-12837-t001:** The index group and weight.

Target Layer	Criterion Layer	Index Layer	Attribute	Weight
Low-carbon development	Low carbon economy	Energy consumption per unit GDP (ton standard coal/10,000 yuan) × 1	Negative	0.1114
		Per capita GDP (10,000 yuan) × 2	Positive	0.1569
	Low-carbon society	Urbanization level (%)× 3	Positive	0.1406
		Ratio of residential land to construction land (%)× 4	Negative	0.0624
	Resource utilization	Water consumption per unit GDP (ton/10,000 yuan) × 5	Negative	0.0745
		Electricity consumption per unit GDP (kilowatt-hour/10,000 yuan) × 6	Negative	0.1307
	Urban planning	Road area per capita (m^2^) × 7	Negative	0.1008
		Number of buses (per 10,000 people) × 8	Positive	0.0970
	Low carbon environment	Per capita green area (m^2^) × 9	Positive	0.1004
		Forest coverage (%)× 10	Positive	0.0253

**Table 2 ijerph-19-12837-t002:** Results of the benchmark model.

Variable	All Cities (282)	Eastern Cities (86)	Central-Western Cities (196)
*ln er*	0.4795 *** (17.260)	0.516 *** (10.790)	0.205 *** (13.483)
*ln ois*	−0.2079	−0.221 ** (−2.120)	0.582
*ln uis*	−0.3068	0.108	−0.541 *** (−3.941)
*ln edu*	0.2173 *** (6.443)	0.107 *** (3.312)	0.227 *** (7.312)
*ln pgdp*	0.2683 *** (6.100)	0.921 ** (2.330)	0.518 *** (9.865)
*ln tec*	0.0659 ** (2.112)	0.114 ** (2.317)	0.156
*ln eol*	0.5099 *** (8.040)	0.432 *** (0.010)	0.244 *** (6.627)
*_cons*	0.2044 *** (35.210)	0.236 *** (14.440)	0.1982 *** (34.474)
*LM test (p)*	4798.27 (0.000)	5038.81 (0.000)	4873.23 (0.000)
*Hausman test (p)*	188.32 (0.000)	149.37 (0.000)	109.18 (0.000)
*Model selection*	individual-fixed effect	individual-fixed effect	individual-fixed effect

Note: **, and *** uniformly indicate significant differences at 5%, and 1% levels, respectively.

**Table 3 ijerph-19-12837-t003:** Global Moran’ I of main variables from 2005 to 2020.

Variable	Global Moran’ I					
	2005	2008	2011	2014	2017	2020
*ulcd*	0.3634 *** (14.77)	0.3752 *** (15.24)	0.3522 *** (14.31)	0.3227 *** (13.13)	0.3125 *** (12.94)	0.3112 *** (12.84)
*er*	0.1221 *** (5.05)	0.2017 *** (15.24)	0.1199 *** (4.98)	0.2124 *** (8.77)	0.1432 *** (6.03)	0.1415 *** (6.04)
*ois*	0.2892 *** (11.82)	0.2557 *** (10.51)	0.2443 *** (10.11)	0.2385 *** (9.75)	0.1890 *** (7.75)	0.2372 *** (9.69)
*uis*	0.1075 *** (5.65)	0.1357 *** (5.69)	0.3522 *** (8.59)	0.2535 *** (10.64)	0.2416 *** (10.07)	0.2341 *** (9.70)

Note: *** indicate significant differences at 1% levels.

**Table 4 ijerph-19-12837-t004:** The spillover effects of ER, OIS, and UIS on Urban low-carbon Development.

Variable	Eastern Cities (86)				Central-Western Cities (196)			
	*SPDM*	*Direct*	*Indirect*	*Total*	*SPDM*	*Direct*	*Indirect*	*Total*
*ln* er	0.508 ***	0.066 ***	0.038 **	0.104 ***	0.270 ***	0.087 **	−0.024 **	0.0634 ***
*ln ois*	−0.306 ***	−0.021 **	−0.072 **	−0.093	0.152	0.055	−0.232	−0.177 *
*ln uis*	0.417	0.056	0.063 **	0.119 **	−0.629 ***	−0.136 ***	−0.042 **	−0.178
*Control variables*	YES	YES	YES	YES	YES	YES	YES	YES
*W*×*ln* er	0.301 **				0.191 **			
*W*×*ln ois*	0.222 ***				−0.312			
*W*×*ln uis*	1.211 **				−0.371 ***			
*W*×*Control variables*	YES	YES	YES	YES	YES	YES	YES	YES
*ρ*	0.235 ***				0.548 ***			
*R* ^2^	0.337				0.394			
*log-likelihood*	1272.313				1339.138			

Note: *, **, and *** uniformly indicate significant differences at 10%, 5% and 1% levels, respectively. (*SPDM: Spatial panel Durbin model; Direct: Direct effect; Indirect: Indirect effect; Total: Total effect*).

**Table 5 ijerph-19-12837-t005:** Test of the threshold effect.

Model	Number of Thresholds	F Value	*p* Value	10% Threshold Level	5% Threshold Level	1% Threshold Level
Eastern cities (86)	1	316.29	0.000	87.779	97.015	113.550
	2	103.29	0.000	44.302	50.652	88.390
	3	19.82	0.960	76.833	88.096	111.203
Central-Western cities (196)	1	616.23	0.000	92.965	116.541	135.959
	2	114.79	0.000	43.985	53.844	67.958
	3	61.34	0.893	136.972	158.408	197.960

**Table 6 ijerph-19-12837-t006:** Estimated results of the environmental regulation threshold.

Model	Threshold Value	95% Confidence Interval	Model	Threshold Value	95% Confidence Interval
Eastern cities (86)	0.5361	(0.5303, 0.5500)	Central-western cities (196)	0.2200	(0.2107, 0.2252)
	0.8110	(0.8090, 0.8120)		0.5834	(0.5802, 0.5876)

**Table 7 ijerph-19-12837-t007:** Estimation results of the panel threshold effect.

Variable	Eastern Cities (86)	Variable	Central-Western Cities (196)
*ln ois* (*er* ≤ 0.5361)	−0.3912 *** (−5.400)	*ln ois* (*er* ≤ 0.2200)	−0.4309 *** (−8.365)
*ln ois* (0.5361 < *er* ≤0.8110)	−0.0135 *** (−7.180)	*ln ois* (0.2200 < *er* ≤0.5834)	−0.0740 *** (−6.511)
*ln ois* (0.8110 < *er*)	−0.0246	*ln ois* (0.5834 < *er*)	0.1362 *** (4.967)
*ln uis* (*er* ≤ 0.5361)	−0.0633	*ln uis* (*er* ≤ 0.2200)	−0.1147 *** (−8.689)
*ln uis* (0.5361 < *er* ≤0.8110)	0.1739 *** (7.770)	*ln uis* (0.2200 < *er* ≤0.5834)	−0.0831 *** (−6.512)
*ln uis* (0.8110 < *er*)	0.1938 *** (7.521)	*ln uis* (0.5834 < *er*)	−0.0111
*control variables*	YES	*control variables*	YES
*_cons*	0.2723 *** (35.497)	*_cons*	0.2208 *** (80.273)
*individual-fixed effect*	YES	*individual-fixed effect*	YES
*F*	243.436 (0.000)	*F test*	245.077 (0.000)

Note: *** indicate significant differences at 1% levels.

**Table 8 ijerph-19-12837-t008:** Estimation results of the robustness test model.

Explanatory Variable: *ln pgCO_2_*	Eastern Cities (86)			Central-Western Cities (196)		
	*BM*	*SPDM*	*PTM*	*BM*	*SPDM*	*PTM*
*ln er*	−0.134 ***	−0.392 ***		−0.223 ***	−0.172 **	
*ln ois*	0.219 ***	0.121 **		0.152	0.138 ***	
*ln uis*	0.037 *	0.176		0.409 ***	0.253 ***	
*Control variables*	YES	YES	YES	YES	YES	YES
*W***ln* er		−0.061 ***			0.085 **	
*W***ln ois*		0.318 **			0.025 *	
*W***ln uis*		−0.115 **			−0.191 ***	
*W*Control variables*		YES			YES	
*ln ois* (*er* ≤ 0.4086) *ln ois* (0.4086 < *er* ≤0.7335) *ln ois* (0.7335 < *er*)			0.2558 *** 0.1152 *** 0.0076 **			
*ln uis* (*er* ≤ 0.4086) *ln uis* (0.4086 < *er* ≤0.7335) *ln uis* (0.7335 < *er*)			0.1745 * −0.2208 *** 0.0994			
*ln ois* (*er* ≤ 0.3427) *ln ois* (0.3427 < *er* ≤0.8834) *ln ois* (0.8834 < *er*)						0.3933 *** 0.2702 *** 0.0820
*ln uis* (*er* ≤ 0.3427) *ln uis* (0.3427 < *er* ≤0.8834) *ln uis* (0.8834 < *er*)						0.4005 *** 0.1118 *** −0.2322 **
*F*	203.343	348.118	289.528	356.718	449.103	395.276
*p*	0.000	0.000	0.000	0.000	0.000	0.000

Note: *, **, and *** uniformly indicate significant differences at 10%, 5%, and 1% levels, respectively.

## Data Availability

Not applicable.

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
