# Peer review of "The Relationship between Environmental Regulation, Industrial Transformation Change and Urban Low-Carbon Development: Evidence from 282 Cities in China"

_ijerph, 2022, doi:10.3390/ijerph191912837_

Round 1

Reviewer 1 Report

This is a fantastic manuscript that delves into the interplay between China's strict environmental regulations, the country's ongoing industrial revolution, and the country's efforts to promote low-carbon urban development. The quality of the paper might be enhanced with a few minor changes, though. Here is a brief synopsis of the feedback:

1.    There are no numerical results presented in the abstract, despite the fact that this is primarily a quantitative study.

2.    What does UIS stand for? Is it an acronym for "upgrading the industrial structure?"

3.    Where is formula A1-A7?

4.   The method section should account for ArcGIS 10.3, which is a prerequisite for its use.

5.   The acronym ULCD should be modified in the legend for figure 1.

6.    Within Section 4.1, titled "Urban Low-Carbon Development Level," there is a need for an additional explanation of ULCD for each region.

Author Response

请看附件

Reviewer 2 Report

This paper is interesting and focus on a hot topic.

I believe the authors shoud explain the reasons selecting the variables.

The data sction should be moved behind the variables. In general, we first decide the variables in the model, then we find the data.

The english writing should be improved.

Reviewer 3 Report

The authors seek to examine effects of environmental regulation (ER) on urban low-carbon development (ULCD) in various parts of China.  I think this is an important topic and relevant to IJERPH. The authors provide sufficient rationale in a three paragraph Introduction.

The second section is the literature review, where the authors do a good job of explaining how the current state of the literature can be brought to bear on policies and results in China.  However, in two places they misspelled the Kuznets curve: line 104 and line 131.  Please correct these mistakes.  They spelled it Kuzlitz in those places.

In section 3 the authors do a good job of laying out material and methods.  They described the number of cities and the years for which they obtained data.  However, I am a little confused by the way the authors describe ER and ULCD.  Throughout the abstract and early part of the manuscript, it looks as though the authors are wanting to examine the effects of ER on ULCD.  This would make ER the explanatory variable and ULCD the dependent variable.  But that is the opposite of what the authors state in lines 153-161.  Furthermore, in regression equations (2) and (3) lines 191 and 205, ULCD appears as the dependent variable.

On lines 215 and 216, the authors return to discussing ULCD as depending on ER, as they do on lines 245 and 246, as well as line 297.  Line 371, at the beginning of the discussion section, also does this as I would expect.

The statistical analysis and its discussion along with the interpretation of results all seem to be fine.  The authors took the effort to deal with potential heteroskedasticity by using a log function.  All of this is well presented.  The authors make extensive use of Tables to present the results. 

The Discussion and Conclusion section are relatively terse and direct, and correctly summarize what the empirical findings showed.  Again though, throughout these sections, the authors write as if the arrow of causality flows from ER to ULCD, making ER the explanatory variable and ULCD the dependent variable. This is the correct way. I am still confused as to why the authors stated the opposite in section 3.2.

I think this manuscript can make a positive contribution.  The authors must correct the spellings on the Kuznets curve and clear up or explain the issues surrounding the dependent variable versus the explanatory variable (ER and ULCD).

Decision: Major revision

Round 2

Reviewer 3 Report

The authors did a great job of revising the manuscript according to the concerns I expressed in my initial review.

In its corrected form, the manuscript should make a useful contribution to the journal.  I like the use of regression they employed.

I think the methods are sound and the conclusions and results flow from the statistical findings the authors obtained.

It was a pleasure to review this manuscript, and I believe that the revisions I suggested have made for a much stronger contribution to the journal.

Decision: Accept